# Multi-granularity Feature Fusion Network for Cross-domain Person Re-identification

Shaoqi Hou[1,2,3,*], Zebang Qin[1], Jiajie Wang[1], Junqi An[1], Yusong Zhang[1], Xinzhong Wang[2], Zhiguo Wang[1,*]
[1]*University of Electronic Science and Technology of China*, Chengdu, China
[2]*Shenzhen Institute of Information Technology*, Shenzhen, China
[3]*Kash Institute of Electronics and Information Industry*, Kashi, China
sqhou@uestc.edu.cn, zgwang@uestc.edu.cn

*Abstract*—In recent years, a lot of encouraging work has emerged in the field of domain generalization as one of the core tools for solving the cross-domain Person Re-identification (Re-ID) task. However, on the one hand, most of the domain generalization methods are devoted to the extraction and aggregation of global or local features, and lack the modeling of structured features of the human body with rich fine-grained information; on the other hand, the existing domain generalization methods emphasize the representation of a single feature, and do not take into full consideration the association and effective fusion of many features. To this end, we propose a Multi-granularity Feature Fusion Network (MFFNet) for cross-domain Re-ID, which utilizes the designed four branching features to construct a complete representation of pedestrians with rich fine-grained and associative information, by drawing on the design idea of graph convolution. Specifically, we effectively construct the intrinsic correlation between local features by introducing the Local Feature Comparison Module (LFCM). In addition, we design a Graph Convolution Module (GCM) to generate pose relational features and topological features with strong discriminative structured information. A series of ablation and comparison experiments on authoritative benchmarks show that the proposed MFFNet achieves competitive performance among similar algorithms.

*Keywords—cross-domain person re-identification, multi-granularity feature fusion network, local feature comparison module, graph convolution module*

## I. INTRODUCTION

Cross-domain Person Re-identification (Re-ID) aims to test the model trained on known source domain data directly on unknown target domain data and expects to achieve good results. In recent years, with the rapid development of deep learning technology and the urgent need to deal with massive surveillance data, cross-domain Re-ID technology has become a core element of intelligent security construction. Although cross-domain Re-ID technology has made some good progress, its actual performance is still difficult to reach the point of ground application, occlusion, lighting, viewing angle, especially background differences, and other complex factors are still the key challenges affecting the performance of cross-domain Re-ID technology.

Due to the significant distributional differences between source and target domain data, how to minimize the "domain difference" has become the core starting point for solving the cross-domain Re-ID task. Since then, many scholars at home and abroad have carried out a lot of substantive exploration around this goal.

The aim of the domain generalization approach is to design a new model that has the ability to mine domain-invariant pedestrian features with strong robustness and discriminative information. Since it does not require the aid of an additional model, this approach is becoming one of the mainstream directions for "domain difference reduction" due to its convenience and efficiency.

At the beginning, a number of algorithms used CNNs and other operations (e.g., HSV histograms, directional gradient histograms, etc.) to extract global features consisting of physical appearance (i.e., color, shape, texture features) [1]. However, extracting only global features inevitably misses some detailed features of pedestrians, so some scholars have focused on the extraction of local features that contain rich fine-grained information. Specifically, Sun et al [2] proposed a PCB network, which divides the pedestrian image into six horizontal chunks to extract different local features. At the same time, they designed an RPP method to enhance the consistency of local features. Guo et al [3] utilized parsing model and self-attention mechanism to extract the features of the human body part and the potential part, respectively, and combined their complementary strengths to enhance the characterization of each pixel. Zheng et al [4] proposed a global-to-local matching model to capture the spatial layout information of the pedestrian images. He et al [5] reconstructed feature maps for local queries from overall pedestrians, and the literature [6] further improved the foreground-background mask to avoid the effect of background clutter. Sun et al [7] proposed a visibility-aware pocalization model (VPM), which extracts region-level features and compares the shared visible regions of two images through self-supervised learning.

In addition, when extracting local features, the local region may contain invalid occluded parts. The re-identification task of occluded pedestrians is more challenging due to incomplete information and spatial misalignment. Zhuo et al [8] proposed a new human attention framework which uses an occlusion simulator to automatically generate artificial occlusions and uses multi-task loss to force the neural network to differentiate between occlusions and full body. Miao et al [9] proposed a new

pose-guided feature alignment method that utilizes pose flags to separate useful information from occlusion noise. Fan et al [10] proposed a spatial-channel parallel network (SCPNet), where the features of each group of channels provide information about a particular spatial region of the human body, and used spatial-channel correlation to train the network. Luo et al [11] used a spatial transformation module to transform the overall image in order to align the local images and then calculated the distance of each set of matched pairs after alignment, in addition to attempting spatial alignment for a local Re-ID task.

However, although the above methods have achieved certain results, they mainly focus on extracting global features or local features, and lack the modeling of structured features of the human body with rich fine-grained information. In addition, the above methods emphasize the expression of individual features and do not fully consider the correlation and effective fusion among many features.

To address the above problems, we design a Multi-granularity Feature Fusion Network (MFFNet), which consists of four branches, including global features, local relational features, gesture-relational features and topological features. The fusion of these branches together constitutes a complete representation of pedestrians with rich fine-grained and correlation information, which effectively improves the robustness and generalization ability of the model on the cross-

domain Re-ID task. In particular, on the one hand, we enable the network to effectively construct the intrinsic correlations among local features by introducing the Local Feature Comparison Module (LFCM) from [12]. On the other hand, we design a Graph Convolution Module (GCM) for generating pose relational features and topological features with strong discriminative structured information by borrowing the idea of higher-order relational and topological modules from [13].

In order to fully demonstrate the effectiveness of the proposed scheme, we conduct complete ablation and comparison experiments on two authoritative benchmarks. The results show that our scheme still achieves competitive performance among similar algorithms even without the aid of additional complex models.

## II. Multi-Granularity Feature Fusion Network

### A. Overall Framework

The overall framework of our MFFNet is shown in Fig. 1, which uses the classical ResNet-50 [14] as the backbone network. It can be seen that MFFNet extracts four kinds of features of pedestrians, including global features, local relational features, gesture relational features and topological features, and finally obtains a complete representation of pedestrians with rich granularity.

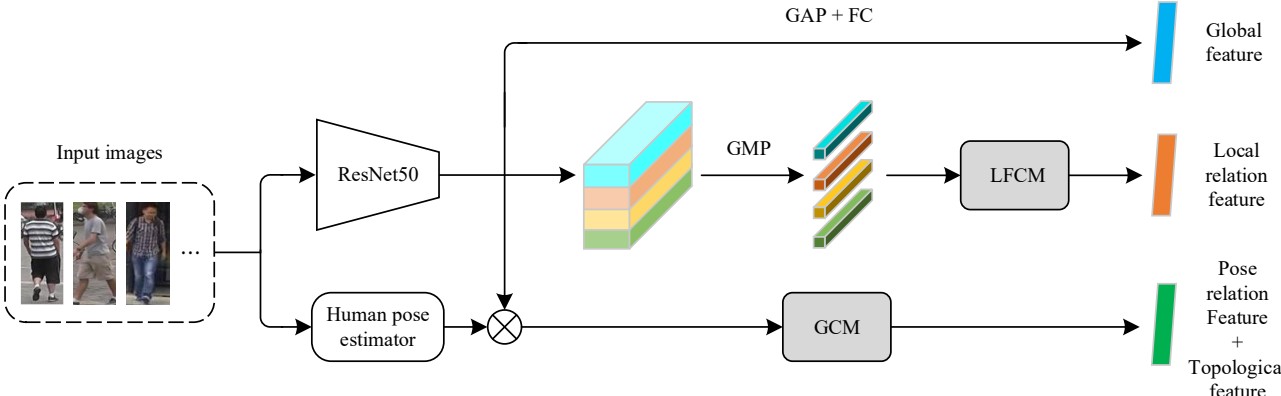

Fig. 1. Overall framework diagram of MFFNet.

Specifically, global feature generalizes the pedestrian image as a whole and pay more attention to the overall feature distribution of pedestrians, but they are easily affected by background clutter and occlusion; local features can well avoid the above problems, and they tend to pay more attention to the local information of pedestrians, such as the upper and lower body. However, supervised training of each local feature individually seems to have a better identification of the query using different local information, but it tends to confuse the local information of the same parts of different pedestrians. Therefore, we introduce the Local Feature Comparison Module (LFCM) from [12] to enable the network to learn the intrinsic correlation between a local feature and other local features, which makes the feature representation of pedestrians more discriminative, as shown in Fig. 2.

In addition, in order to further alleviate the occlusion problem as well as extract pedestrian features with strong

robustness and discrimination, we introduce the higher-order relation module and topology module from [13] and combine them into the proposed Graph Convolution Module (GCM) (shown in Fig. 3), and the final outputs are obtained as the gesture relation features and topology features. The graph convolution technique can make full use of the intrinsic correlations between points and points, edges and edges to enable the network to learn effective and robust pedestrian feature representations, which on the one hand can better cope with the occlusion problem, and on the other hand enhances the final and complete representation of pedestrian features. Specifically, since we need to obtain the pedestrian's pose features before performing graph convolution, we use a good human pose estimator, LIP-JPPNet [15], which can predict the positions of 16 key points of the human body, namely, the head, neck, spine, and pelvis individually as well as the shoulders, elbows, wrists, hips, knees, and ankles in pairs.

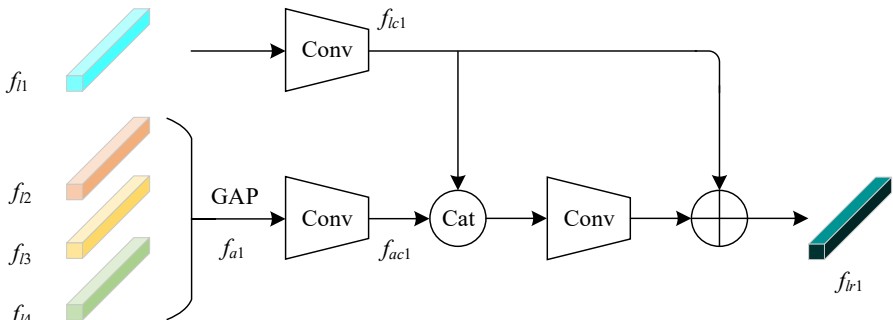

Fig. 2.  Structure of LFCM.

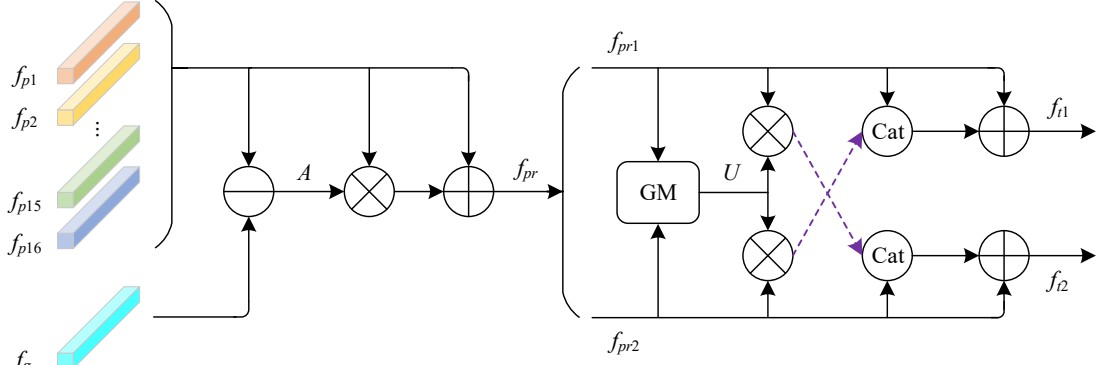

Fig. 3.  Structure of GCM.

In MFFNet, considering that a large number of local features (e.g., horizontal division and human posture) have been extracted and correlation learning has been carried out, in order to prevent network redundancy and learning conflicts, we do not carry out correlation processing on global features, and we obtain the pedestrian global features only by performing the global average pooling and fully connected operation on the features output from the backbone network. Finally, the global feature, local relational features, gesture relational features and topological features are fused to obtain the complete pedestrian feature representation of the proposed framework, which significantly improves the domain generalization ability of the model.

*B.  Algorithm Flow*

As shown in Fig. 1, the processing flow of MFFNet is as follows:

Step 1: The input image is processed by ResNet-50 to get the original feature map $f_o$ with dimension C×H×W.

Step 2: Global feature acquisition. Global average pooling (which pays more attention to the overall style of the image) and fully connected operations are performed sequentially on the original feature map $f_o$ to obtain the global feature $f_g$ with dimension 1×1×C.

Step 3: Acquisition of local relational features. First, the original feature map $f_o$ is partitioned into 4 regions in the horizontal direction on average to obtain 4 feature maps with dimensions C×H/4×W. Then after global maximum pooling (which pays more attention to the saliency information of the

whole region) is performed for each block region to obtain 4 local features $f_{l1}$, $f_{l2}$, $f_{l3}$ and $f_{l4}$ with dimensions 1×1×C respectively.

Then, the four local features are inputted into the LFCM for processing to obtain the local relational features $f_{l1}$, $f_{l2}$, $f_{l3}$ and $f_{l4}$, respectively. Fig. 2 illustrates the computation process of $f_{l1}$, and the other local relational features are computed in the same way.

As we know, in the past, local features were usually processed by stitching them together in a specific order and representing them as a structured form with a certain meaning. This approach is robust to the variation of pedestrian data, however, in real scenarios the data may be more diverse and difficult to learn, only limited information about pedestrians can be extracted using these features alone. Moreover, this approach does not exploit the rich correlation information hidden among many local features, and does not fully utilize the identification performance of the network. Therefore, we introduce LFCM to effectively alleviate this problem, so that each local feature contains both its own information and the information of other local features.

Specifically, taking the computation of the local relational feature $f_{l1}$ as an example, Global Average Pooling (GAP) is first performed on the other local features $f_{l2}$, $f_{l3}$ and $f_{l4}$ to obtain the aggregated feature $f_{a1}$:

$$f_{a1} = \frac{1}{3}\sum_{j\neq 1} f_{lj} \qquad (1)$$

Then, 1×1 convolution is performed on $f_{l1}$ and $f_{a1}$ respectively to compress the number of channels to c, obtaining feature maps $f_{lc1}$ and $f_{ac1}$, both with dimensions 1×1×c. Next, $f_{lc1}$ and $f_{ac1}$ are channel spliced (the number of channels becomes 2c), and then processed by a 1×1 convolution unit (with the addition of the batch normalization and the ReLU function) to reduce the dimension from 2c to c. Finally, the the feature after channel number reduction is summed with $f_{lc1}$, the local relational feature $f_{lr1}$ containing both $f_{l1}$ itself and other local information is obtained:

$$f_{lr1} = f_{lc1} + \text{Conv}(\text{Cat}(f_{lc1}, f_{ac1})) \qquad (2)$$

where Cat(·) denotes the splicing operation and Conv(·) denotes the 1×1 convolutional unit. In particular, the other local relational features $f_{lr2}, f_{lr3}$ and $f_{lr4}$ can also be computed from Eq. (1) and Eq. (2).

It should be emphasized that the loss functions used in the training of both the local relational features here and the two branches of the global relational features described above are the standard cross entropy loss and the triplet loss.

Step 4: Acquisition of pose relational features and topological features. First, the input image is processed by the human posture estimator to obtain a feature map $f_k$ containing 16 keypoint coordinates, and $f_k$ is multiplied with the original feature map $f_o$ to obtain 16 posture features $f_{p1}, f_{p2} \cdots f_{p15}$ and $f_{p16}$ with dimensions 1×1×C. Then, the posture features and the above global features $f_g$ are inputted into the GCM for processing, which is used to learn the pose relationship and topological relationship among different pedestrians.

As shown in Fig. 3, the GCM operates as follows:

1) The difference between the global feature $f_g$ and a set of pose features $f_p$ (i.e., $f_{p1}, f_{p2} \cdots f_{p16}$) is computed to obtain the weights of the edges between the pose points (which are used to characterize the correlation between the points), denoted as $A$.

2) $A$ is multiplied by $f_p$, and then the obtained result is summed with $f_p$ to obtain the pose relational feature $f_{pr}$:

$$f_{pr} = A^{adp} \times f_p + f_p \qquad (3)$$

The loss function for $f_{pr}$ still uses a combination of cross entropy loss (denoted by $L_{cls}$) and triplet loss (denoted by $L_{tri}$) with the following formula:

$$
\begin{aligned}
L_{pr} &= \frac{1}{K} \sum_{k=1}^{K} \beta_k [L_{cls} + L_{tri}] \\
&= \frac{1}{K} \sum_{k=1}^{K} \beta_k [-\log p_{pr} + \max\{\alpha + d_{a,p} - d_{a,n}, 0\}]
\end{aligned} \qquad (4)
$$

where $K$=16 and $\beta_k$ is the confidence level of the $k$-th key point (range is the interval [0,1]); $p_{pr}$ denotes the probability of predicting that $f_{pr}$ belongs to the real identity; $\alpha$ is the boundary value, $(a,p)$ denotes the positive sample feature pairs from the

same identity, and $(a,n)$ denotes the negative sample feature pairs from different identities.

3) Given two input images $i_1$ and $i_2$, follow the above steps to compute the two sets of pose relational features $f_{pr1}$ and $f_{pr2}$, respectively. the goal of Graph Matching (GM) is to learn the matching matrix $U$ (of dimension $K \times K$ and values between 0 and 1) between $f_{pr1}$ and $f_{pr2}$, where the element $U_{ij}$ denotes the degree of match between the $i$-th point in $f_{pr1}$ and the $j$-th point in $f_{pr2}$.

4) Define a symmetric matrix $M$ (of dimension $K^2 \times K^2$), where the elements $M_{ac;bd}$ are used to measure the match between the edge $(a,b)$ in $f_{pr1}$ and the edge $(c,d)$ in $f_{pr2}$, and the pairs of points that do not form an edge are set 0 in the matrix. Therefore, the optimal matching $U^*$ is defined as Eq. (5), and we use the characteristics of the pairs of points to initialize $M$, and optimize $U$ by a stochastic gradient descent algorithm.

$$U^* = \underset{U}{arg\,gm\,ax} U^T M U \qquad (5)$$

5) After obtaining the matching matrix $U$, a set of topological features $f_{t1}$ and $f_{t2}$ are finally obtained by multiplication, splicing, convolution, and summation operations shown in Fig. 3:

$$
\begin{aligned}
f_{t1} &= f_{pr1} + \text{Conv}(\text{Cat}(f_{pr1}, f_{pr2} \times U)) \\
f_{t2} &= f_{pr2} + \text{Conv}(\text{Cat}(f_{pr2}, f_{pr1} \times U))
\end{aligned} \qquad (6)
$$

The loss function of the resulting topological features is shown in Eq. (7), $y$=1 when $i_1$ and $i_2$ are from the same pedestrian, otherwise $y$=0.

$$L_t = y \log |f_{t1} - f_{t2}| + (1-y) \log(1 - |f_{t1} - f_{t2}|) \qquad (7)$$

## III. EXPERIMENTS AND ANALYSIS

### A. Implementation Details

**Datasets.** Our experiments use two of the most commonly used public datasets in the Re-ID field, Market1501 [16] and DukeMTMC-reID [17], whose details are shown in TABLE I. Specifically, M→D denotes training on Market1501 and testing on DukeMTMC-reID, while D→M does the opposite.

**Evaluation metrics.** All experiments were evaluated using the Cumulative Match Characteristic (CMC) and mean Average Precision (mAP) for Rank-1, Rank-5, and Rank-10, which are described in detail in the literature [18].

**Experimental setup.** The experimental environment we used is Python3.6+Torch1.4+Torchvision0.5+CUDA11.2, and the other experimental settings are the same as in the literature [12,13]. For training, all input images are Resize to 256×128 (C and c mentioned above are set to 2048 and 256, respectively), and random horizontal flipping, random cropping, and random erasure are performed to enhance the diversity of the data. The Batch for each iteration is set to 64 (16 identities in total, with 4 images selected for each identity). In addition, the gradient update is performed using the Adam optimizer, with the initial learning rate set to 0.001 and reduced to 0.1 times the previous

one after every 20 epochs of training, stopping after a total of 100 epochs of training. In particular, the total loss of this model is the weighted sum of the losses of each feature branch, and the weight parameter is set to 1 by default.

| Datasets | Train | | Query | | Gallery | |
|---|---|---|---|---|---|---|
| | *IDs* | *Images* | *IDs* | *Images* | *IDs* | *Images* |
| Market1501 | 751 | 12,936 | 750 | 3,368 | 750 | 19,732 |
| DukeMTMC-reID | 702 | 16,522 | 702 | 2,228 | 702 | 17,661 |

## B. Ablation Experiments and Analysis

### 1) Effect of number of horizontal chunks on algorithm performance

The partitioning of different horizontal regions has an important impact on the computation of local relational features and their fusion with global features, pose relational features, and topological features. In order to explore the optimal number of chunks that can maximize the effect of GCM, we compute the local relational features and fuse them with other features on 2, 3, 4, 5 and 6 horizontal regions (i.e., average segmentation according to H/2, H/3, H/4, H/5 and H/6 in H dimension, respectively) and supervise the training of each region individually in each experiment, and the experimental results are shown in TABLE II.

TABLE II.      EFFECT OF NUMBER OF HORIZONTAL CHUNKS ON ALGORITHM PERFORMANCE

| Nums | M → D (%) | | | | D → M (%) | | | |
|---|---|---|---|---|---|---|---|---|
| | *Rank-1* | *Rank-5* | *Rank-10* | *mAP* | *Rank-1* | *Rank-5* | *Rank-10* | *mAP* |
| 2 | 66.4 | 75.2 | 78.4 | 52.1 | 79.2 | 85.6 | 88.6 | 58.2 |
| 3 | 68.2 | 77.5 | 80.6 | 54.7 | 81.1 | 87.4 | 90.4 | 60.3 |
| 4 | **69.1** | **78.2** | **81.4** | **55.9** | **82.0** | **88.5** | **91.7** | **61.8** |
| 5 | 68.5 | 77.8 | 80.9 | 54.8 | 81.3 | 87.7 | 90.4 | 60.5 |
| 6 | 66.3 | 75.0 | 78.1 | 52.0 | 79.1 | 85.5 | 88.7 | 58.1 |

As can be seen from TABLE II, the performance of the proposed algorithm shows an increasing and then decreasing trend with the increase of the number of chunks. When the number of horizontal chunks is small, the performance of MFFNet is poor, which may be due to the fact that the local relational features derived at this time do not have more local characteristics, and even conflict with the global features, which can easily make the learning of the network misguided. When the number of horizontal chunks is higher, the metrics Rank-1 and mAP also undergo a significant decline, for example, when cross-domain identification is performed on M→D, the Rank-1 and mAP of the division of 6 chunks are 2.2% and 2.8% lower than those of the division of 5 chunks, respectively. It is sufficient to show that more chunks are not better, although more chunks may make the network more capable of distinguishing different pedestrians based on delicate local features, but it may lead to the conflict of multiple learning tasks as numerous pose features and topological features are also introduced in this paper. In addition, the more chunks are divided, the greater the number of parameters and computational effort of the overall model. Therefore, the 4-chunks horizontal chunks division is more suitable for the proposed network structure and enables the model to perform the best cross-domain identification performance, which will be consistent in subsequent experiments.

### 2) Impact of fusion of different features on algorithm performance

Our MFFNet extracts global features, local relational features, pose relational features and topological features of pedestrian images and uses these features to obtain a complete feature representation of pedestrians with rich fine-grain and strong discriminative power, which gives the model strong robustness and generalization ability. In order to verify the effect of the fusion of different features mentioned above on the performance of cross-domain Re-ID, we have done a large number of ablation experiments, and the results are shown in TABLE III.

From TABLE III, it can be seen that the worst re-identification performance is obtained by using only global features for training and testing. After adding local relational and topological features to the global features, respectively, the Rank-1 and mAP of the proposed model went up by about 4 percentage points. Further, the introduction of both local relational and topological features brings about a cumulative performance gain (more than 8%), which is enough to show that modeling the intrinsic associations between local features and aligned matching of human postures are necessary for the re-identification task. In addition, after adding the pose relational features to the global features, the model's re-identification effect is remarkable, with the metrics Rank-1 and mAP

improving by about 12% for both the M→D task and the D→M task, which reflects sideways the robustness of the pose relational features to the most common occlusion problems in pedestrian images. Undoubtedly, the model after fusing the four features establishes a richer and more complete pedestrian feature representation with the addition of multi-granularity information, and also achieves the best re-identification performance.

TABLE III.    IMPACT OF FUSION OF DIFFERENT FEATURES ON ALGORITHM PERFORMANCE

| $f_g$ | $f_{lr}$ | $f_{pr}$ | $f_t$ | M → D (%) | | | | D → M (%) | | | |
|---|---|---|---|---|---|---|---|---|---|---|---|
| | | | | *Rank-1* | *Rank-5* | *Rank-10* | *mAP* | *Rank-1* | *Rank-5* | *Rank-10* | *mAP* |
| √ | | | | 49.0 | 57.8 | 62.1 | 35.2 | 61.8 | 65.9 | 70.7 | 40.4 |
| √ | √ | | | 53.2 | 61.5 | 66.4 | 39.3 | 65.6 | 69.7 | 74.4 | 44.5 |
| √ | | √ | | 61.0 | 69.9 | 74.3 | 47.4 | 73.6 | 79.4 | 82.9 | 52.4 |
| √ | | | √ | 53.8 | 61.4 | 66.7 | 39.5 | 65.0 | 69.1 | 74.6 | 44.7 |
| √ | √ | √ | | 65.3 | 73.2 | 78.5 | 51.8 | 77.7 | 83.7 | 86.3 | 56.2 |
| √ | √ | | √ | 57.1 | 65.4 | 70.8 | 43.0 | 69.7 | 73.1 | 78.9 | 48.5 |
| √ | | √ | √ | 65.7 | 73.4 | 78.9 | 51.4 | 77.4 | 83.0 | 86.0 | 56.8 |
| √ | √ | √ | √ | **69.1** | **78.2** | **81.4** | **55.9** | **82.0** | **88.5** | **91.7** | **61.8** |

## C. Comparison with state-of-the-art algorithms

In order to demonstrate the superiority of the proposed MFFNet in terms of overall modeling, we have selected some state-of-the-art algorithms in recent years for comparison, including PUL [19], TJ-AIDL [20], and MMFA [21], and a total of seven cutting-edge works.

As can be seen from TABLE IV, the metrics of MFFNet have significant advantages over most of the selected algorithms, especially the most critical Rank-1 and mAP achieve the best performance on both M→D and D→M tasks. Specifically, compared with the best UDAP, our MFFNet exceeds the Rank-1 metrics by 0.7% and 6.2%, and the mAP metrics by 6.9% and 8.1% for the two cross-domain tasks, respectively. It is fully demonstrated that our mining, construction and fusion of global features, local relational features, pose relational features and topological features have accomplished an accurate description of the discriminative representation of pedestrian images in cross-domain scenes.

TABLE IV.    COMPARISON WITH STATE-OF-THE-ART ALGORITHMS

| Models | M → D (%) | | | | D → M (%) | | | |
|---|---|---|---|---|---|---|---|---|
| | *Rank-1* | *Rank-5* | *Rank-10* | *mAP* | *Rank-1* | *Rank-5* | *Rank-10* | *mAP* |
| PUL [19] | 30.0 | 43.4 | 48.5 | 16.4 | 45.5 | 60.7 | 66.7 | 20.5 |
| TJ-AIDL [20] | 44.3 | 59.6 | 65.0 | 23.0 | 58.2 | 74.8 | 81.1 | 26.5 |
| MMFA [21] | 45.3 | 59.8 | 66.3 | 24.7 | 56.7 | 75.0 | 81.8 | 27.4 |
| HHL [22] | 46.9 | 61.0 | 66.7 | 27.2 | 62.2 | 78.8 | 84.0 | 31.4 |
| BUC [23] | 47.4 | 62.6 | 68.4 | 27.5 | 66.2 | 79.6 | 84.5 | 38.3 |
| ECN [24] | 63.3 | 75.8 | 80.4 | 40.4 | 75.1 | 87.6 | 91.6 | 43.0 |
| UDAP [25] | 68.4 | **80.1** | **83.5** | 49.0 | 75.8 | **89.5** | **93.2** | 53.7 |
| MFFNet (Ours) | **69.1** | 78.2 | 81.4 | **55.9** | **82.0** | 88.5 | 91.7 | **61.8** |

## IV. CONCLUSIONS

We propose a multi-granularity feature fusion network (MFFNet) to enhance the robustness and generalization ability of the model on cross-domain Re-ID tasks. Different from the previous focus on processing a single feature descriptor, MFFNet effectively fuses multiple descriptors such as global features, local relational features, pose relational features, and topological features, which together form a complete representation of pedestrians with rich fine-grained and correlation information. To deal with the most challenging pedestrian occlusion problem, we draw on the principles of graph convolution operation and design a novel graph convolution module, GCM.Encouragingly, GCM dynamically

controls the direction and degree of discriminative message passing by modeling the relationships between the pedestrian skeleton features point-to-point and edge-to-edge, and guiding the network to continuously learn the pose characteristics and topological features of the pedestrians in the image.The effectiveness of the proposed method is demonstrated by testing the two cross-domain tasks M→D and D→M. In future work, we will try to introduce the image style transformation method based on MFFNet, with the aim of designing a better cross-domain Re-ID scheme from both the model construction and data processing perspectives at the same time.

## ACKNOWLEDGMENT

This work was supported by the Natural Science Foundation of Xinjiang Uygur Autonomous Region under grant number 2022D01B186.

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
