# OpenReview forum: "Multi-granularity Feature Fusion Network for Cross-domain Person Re-identification"
_IEEE.org/ICIST/2024/Conference — IEEE ICIST 2024 Conference Submission_

### Official Review · Reviewer_tAbL · 2024-08-22
**accept**

**Rating:** 8
**Confidence:** 4

**Review:**

This paper proposes a Multi-granularity Feature Fusion Network (MFFNet) for cross-domain Re-ID, which utilizes the designed four branching features to construct a complete representation of pedestrians with rich fine-grained and associative information, by drawing on the design idea of graph convolution. A series of ablation and comparison experiments on authoritative benchmarks show that the proposed MFFNet achieves competitive performance among similar algorithms.
1). In abstract, about the proposed method, the statement is unclear. Authors need to rewrite abstract and to focus on the proposed method and to stress both the specific application and the generic aspects of the work.
2). The paper needs to clearly indicate its contributions to show the superiority of the proposed fuzzy approach on the previous approach.

---

### Official Review · Reviewer_U6qj · 2024-08-24
**The paper is written clearly, exceptionally excellent.**

**Rating:** 8
**Confidence:** 3

**Review:**

This paper excels in terms of quality, clarity, originality, and significance, but I would still like to offer some suggestions.
1. Emphasize the original work of the paper.
2. Further emphasize the research motivation.

---

### Official Review · Reviewer_JJu4 · 2024-08-25
**Current domain generalization techniques often focus on extracting and combining either global or local features but neglect the detailed modeling of structured human body features with fine-grained information. They also tend to prioritize individual feature representation without considering feature associations and effective fusion. To address these limitations, we introduce the Multi-granularity Feature Fusion Network (MFFNet) for cross-domain Re-ID tasks. MFFNet employs four distinct branching features to create a comprehensive pedestrian representation that includes fine-grained details and associative data. It incorporates the Local Feature Comparison Module (LFCM) for establishing relationships among local features and the Graph Convolution Module (GCM) for generating pose relational and topological features. Rigorous ablation studies and benchmark comparisons show that MFFNet performs competitively with other state-of-the-art algorithms.But they needExplain what assumptions and lemma are used, and explain the rationale. Originality needs to be further emphasized.**

**Rating:** 7
**Confidence:** 4

**Review:**

In order to solve most domain generalization techniques currently focus on extracting and combining either global or local features, overlooking the modeling of structured human body features that contain rich, fine-grained information. Furthermore, existing methods tend to concentrate on the representation of individual features, neglecting the importance of feature associations and effective fusion. To tackle these issues, we introduce the Multi-granularity Feature Fusion Network (MFFNet) tailored for cross-domain Re-ID tasks. This novel network harnesses four distinct branching features to craft a comprehensive pedestrian representation enriched with fine-grained details and associative data, inspired by graph convolution principles. Specifically, we’ve incorporated the Local Feature Comparison Module (LFCM) to efficiently establish the inherent relationships among local features. Additionally, we’ve devised a Graph Convolution Module (GCM) aimed at generating pose relational and topological features, packed with highly discriminatory structured information. Rigorous ablation studies and comparative experiments on established benchmarks demonstrate that our proposed MFFNet offers competitive performance among similar algorithms.
1.However,Originality needs to be further emphasized.
2.Explain what assumptions and lemma are used, and explain the rationale.

---

### Decision · Program_Chairs · 2024-09-08

Accept (Oral)